# Exciton Dynamics in Droplet Epitaxial Quantum Dots Grown on (311)A-Oriented Substrates

**DOI:** 10.3390/nano10091833

**Published:** 2020-09-14

**Authors:** Marco Abbarchi, Takaaki Mano, Takashi Kuroda, Kazuaki Sakoda

**Affiliations:** 1Aix Marseille University, Université de Toulon, CNRS, IM2NP Marseille, France; 2Research Center for Functional Materials, National Institute for Materials Science, 1-1 Namiki, Tsukuba, Ibaraki 305-0044, Japan; MANO.Takaaki@nims.go.jp (T.M.); KURODA.Takashi@nims.go.jp (T.K.); SAKODA.Kazuaki@nims.go.jp (K.S.)

**Keywords:** III-V quantum dots, droplet epitaxy, exciton dynamics, (311)A oriented substrate

## Abstract

Droplet epitaxy allows the efficient fabrication of a plethora of 3D, III–V-based nanostructures on different crystalline orientations. Quantum dots grown on a (311)A-oriented surface are obtained with record surface density, with or without a wetting layer. These are appealing features for quantum dot lasing, thanks to the large density of quantum emitters and a truly 3D lateral confinement. However, the intimate photophysics of this class of nanostructures has not yet been investigated. Here, we address the main optical and electronic properties of s-shell excitons in individual quantum dots grown on (311)A substrates with photoluminescence spectroscopy experiments. We show the presence of neutral exciton and biexciton as well as positive and negative charged excitons. We investigate the origins of spectral broadening, identifying them in spectral diffusion at low temperature and phonon interaction at higher temperature, the presence of fine interactions between electron and hole spin, and a relevant heavy-hole/light-hole mixing. We interpret the level filling with a simple Poissonian model reproducing the power excitation dependence of the s-shell excitons. These results are relevant for the further improvement of this class of quantum emitters and their exploitation as single-photon sources for low-density samples as well as for efficient lasers for high-density samples.

## 1. Introduction

Low-dimensional nanostructures [1] have been widely employed for a plethora of applications ranging from sensing [2], light harvesting in solar cells [3], biology [4], and photonics [5,6,7,8,9,10]. Among the different categories, epitaxial quantum dots (QDs) stand as the best alternative for quantum devices thanks to their brightness, stability, and compatibility with photonic and electronic devices [7,8,10,11,12]. Within this class of QDs, droplet epitaxy (DE) [10,13,14] and droplet etching [9,15,16] (alternative growth protocols to Stranski–Krastanov for strain-free III–V-based semiconductor nanostructures), enabled the fabrication of state-of-the-art devices such as lasers [17,18,19,20,21] and quantum emitters, including single-photon sources [22,23,24,25,26] and entangled photons [9,27,28,29,30] with electrical injection [31]. The versatility of this method allowed to grow many different semiconductor alloys (GaInSb [32], AlGaAs [33,34,35,36,37], InGaAs [38,39,40,41,42,43,44], and InGaP [26,45,46]), forming a plethora of nanostructures [47] such as quantum dots (QDs); QDs diads [48,49,50]; multiple-concentric quantum rings [18,23,51,52,53,54,55,56]; coupled structures such as ring-on-a-disk [57], dot in-a-ring [58], or dot-on-a-disk [59]; as well as elongated structures such as nanowires [60]. This technique allows to independently tune the size and density of the nanostructures [61] and to grow them with or without a wetting layer [35,39,40,60,62,63,64,65,66], aspects that are not matched by the conventional Stranski–Krastanov approach based on strain [67].

Another aspect relevant for applications is the possibility to grow high-quality nanostructures on different substrate orientation, providing the ground for highly symmetric QDs on (111)A surfaces [26,27,35,36,41,42,46,61,68,69,70,71] (e.g., for entangled photon pairs generation) and for ultra-high density quantum wires [60] and QDs [19,32,38,63,64,65,72,73] formation on the highly anisotropic (311)A surface (e.g., for laser emission). This latter class of nanostructures grown on (311)A surface has not yet been thoroughly investigated and a clear assessment of the corresponding excitonic dynamics has not yet been reported.

In this paper, we show a detailed structural and optical characterization of individual QDs grown on the (311)A surface. We provide a clear-cut attribution of the main recombination lines observed in the photoluminescence (PL) spectrum to the s-shell excitons [33,73,74] based on polarization-resolved PL measurements, power dependence, and line broadening measurements. Neutral exciton X and biexciton XX, positive X${}^{+}$ and negative charged excitons X${}^{-}$ are characterized by the presence of fine interactions [33,34,40,75] as well as by heavy-hole/light-hole mixing [76,77,78,79,80]. Their power dependence under laser excitation above barrier is well reproduced by a simple Poissonian model that precisely accounts for the main features and allows to estimate the excitonic capture volume [81]. Inhomogeneous line broadening at low temperature is ascribed to spectral diffusion [24,30,82,83,84,85] induced by the presence of charged defects nearby the QDs [86] and is specific to the excitonic complex in study [87,88]. Finally, at higher temperature, the photoluminescence of individual QDs broadens and quenches owing to phonon interactions [89,90,91].

## 2. Materials and Methods

### 2.1. Sample Fabrication

The sample was grown on a semi-insulating GaAs (311)A substrate by conventional solid-source molecular-beam epitaxy system (MBE32 by Riber). After the growth of a 2 μm thick Al${}_{0.55}$Ga${}_{0.45}$As layer, a 136 nm thick Al${}_{0.26}$Ga${}_{0.74}$As core-layer was grown at 610 ${}^{\circ }$C. At the core layer center, GaAs QDs were formed by droplet epitaxy. On the Al${}_{0.26}$Ga${}_{0.74}$As surface, nominally 1.5 monolayers of Ga (at a growth speed of ~0.1 monolayers per second) was supplied without As${}_{4}$ flux at 275 ${}^{\circ }$C for the droplets formation. These droplets were then crystallized into GaAs QDs by supplying an As${}_{4}$ flux (2 × 10${}^{-6}$ Torr beam equivalent pressure) at 200 ${}^{\circ }$C. The QDs were annealed at 400 ${}^{\circ }$C for 10 min under As${}_{4}$ flux supply without capping in order to improve the crystal quality. After annealing, the QDs were covered with a 30 nm thick Al${}_{0.26}$Ga${}_{0.74}$As capping layer at 400 ${}^{\circ }$C and the rest of the Al${}_{0.26}$Ga${}_{0.74}$As (38 nm) layer was grown at 625 ${}^{\circ }$C.

Once the entire growth sequence was completed, a rapid thermal annealing process was performed at 785 ${}^{\circ }$C for 4 min in an As${}_{4}$ atmosphere to improve the optical quality. The effect of ex situ rapid thermal annealing step has been reported in [18,92]. From these studies it emerges that nanostructures crystallized into GaAs by a supply of low As flux show a dramatic increase of the corresponding emission intensity. Transmission electron microscopy [18] shows a negligible effect on the nanostructure morphology with a very small thickening of the wetting layer. In spite of the rather small interdiffusion coefficient between GaAs and AlAs [93], for high-temperature post-growth annealing, a modification of the PL band with a blue shift can be observed [92].

In order to characterize the QDs density and morphology, a similar sample was grown without capping, thus leaving the QDs exposed for otimic force microscopy investigation. For that purpose we used an atomic force microscope (AFM, SPA400 by Hitachi High-Tech) in non-contact mode.

### 2.2. Optical Spectroscopy

The photoluminescence (PL) of individual QDs was collected with a confocal-spectroscopic set-up (lateral resolution of ~1 μm) CW laser excitation was performed above-barrier, at 532 nm (~2.3 eV). The PL is then fed into a spectrometer and detected by a Si-based charge-coupled device (CCD) camera, allowing for a spectral resolution better than 50 μeV in full width at half maximum (FWHM). All experiments were performed in a liquid helium cryostat between 10 and 100 K.

## 3. Results and Discussion

We first provide a structural characterization of (311)A QDs via AFM study (Figure 1) in order to highlight their strongly anisotropic shape. Figure 1a,b shows atomic force micrographs (AFM) of the sample after annealing at 400 ${}^{\circ }$C. Well-defined QDs are present with a density of approximately 5 × 10${}^{9}$/cm${}^{2}$. The QD morphology is highly asymmetric, with a U-shape (Figure 1b). We attribute the formation mechanism of these asymmetric QDs to the low-intensity As${}_{4}$ supply for the crystallization of the droplets and surface asymmetry of the (311)A surface [19,32,38,60,63,64,65,72]. In this growth condition, the Ga droplets crystallization into GaAs is enhanced around the edge of the droplets. In the case of standard (100) surface, in which the oppositely oriented directions of [011] and [0-11] are equivalent, ring-like structures with central holes are formed [51,52]. On the (311)A surface, however, the oppositely oriented directions of [-233] and [2-3-3] are not equivalent, while the directions of [01-1] and [0-11] are equivalent. Thus, the crystallization is enhanced only in a particular direction ([-233]). As a result, U-shaped QDs are formed.

Macro PL on the QDs ensemble taken at low temperature shows the different contributions from the GaAs substrate (at ~1.48 eV), the Al${}_{0.26}$Ga${}_{0.74}$As barriers (at ~1.85 eV) and the QDs in between, extending over a broad band from approximately 1.55 eV up to 1.75 eV. Aside from this, we note a small PL peak at ~1.82 eV. We attribute this emission to shallow centers due to carbon-related residual contamination (donor–acceptor carbon (DA-C)) that is commonly found in epitaxial QDs grown in molecular beam epitaxy reactors [94,95].

### 3.1. s-Shell Excitons

Typical spectra of individual QDs on (311)A substrate appear structured in several sharp lines (Figure 2a). Although at the center of the macro-PL band (at approximately 1.62 eV) the spatial density of QDs is larger making the isolation of individual QDs more complicate, emission from single QDs in the full range from about 1.6 up to 1.8 eV is in principle possible [73]. In analogy to most III-V epitaxial QDs, including those fabricated via DE [22,33,73,96], in the low excitation power regime, the s-shell excitons dominate the PL spectrum. We ascribe the brightest lines to the recombination of the neutral exciton X (one electron, *e*, and a heavy-hole, *hh*), neutral biexciton (XX, two *e* and two *hh*), positive charged exciton (one *e* and two *hh*), and to the negative charged exciton (two *e* and a *hh*). This attribution is based on the study of the electron–hole fine interaction [33] (fine structure splitting), line broadening [82] (spectral diffusion), and level filling [81], as discussed in the following sections. Seldom have we also observed other sharp lines, such as M0 and M1 (Figure 2a), that likely involve carrier recombination from the p-shell. However, a precise attribution of these lines is not possible with this set of data and goes beyond the aim of this paper.

### 3.2. *e-h* Spin Fine Interaction: Fine Structure Splitting

A first identification of the s-shell excitons is based on the study of the fine interaction between electron and *hh* spin [33,34,75,97] (the fine structure splitting (FSS), Figure 2a)). At intermediate laser excitation power (at approximately 420 nW) all the PL components in the spectrum are well visible (Figure 2a). At this power, by changing the polarization angle of the detected light, we monitor the emission energy of the PL. We observe for X and XX a mirror-symmetric energy splitting (Figure 2c). X${}^{+}$ and X${}^{-}$ instead, do not feature any splitting.

This picture corresponds very well to what is commonly observed in strain-free, III-V QDs: the presence of geometrical asymmetries in the confining potential breaks the invariance of the Hamiltonian for rotations around the vertical growth axis [33,34,37,75,78,97,98]. More precisely, *e-h* spin–spin interaction for the X state lifts the degeneracy of the corresponding energy level and leads to two linearly polarized PL components, depending on which recombination path is radiating. The XX state has overall zero spin for *e* and *hh* in the initial state and its FSS is completely determined by that of the X state, towards which it relaxes. X${}^{+}$ and X${}^{-}$ can recombine with two possible energy equivalent paths and have no FSS.

The measured FSS in this QD is of ~100 μeV. In other QDs we measured values between 30 and 140 μeV (not shown). These values are very large if compared with QDs grown on (111) substrates, where the three-fold symmetry of the crystal provides a more isotropic surface diffusion and a corresponding triangular or hexagonal nanostructure shape [26,27,35,36,41,42,46,61,68,69,70,71]. The origin of this splitting in (311)A QDs is attributed to asymmetries in the QDs shape that is affected by the anisotropy of the underlying crystal. The measured values of FSS are in line with those observed in (001) QDs in spite of the larger anisotropy of the (311)A surface [33,34,75,97]. Owing to the growth method in use, that allows for 3D growth of lattice-matched materials such as GaAs on AlGaAs, we exclude the presence of any strain in the structure that is a major origin of symmetry breaking (and thus FSS) in conventional Stranski–Krastanov III-V QDs [97].

### 3.3. Polarization Intensity Anisotropy: Heavy-Hole Light-Hole Mixing

We monitor the polarization anisotropy of X and XX split components observing the different intensity of the corresponding PL lines (Figure 2c)). A remarkable intensity change is visible for the two orthogonal polarized components (between 1/3 and 1/2, depending on the observed line, Figure 2c)) of X and XX as well as in the polarization intensity of X${}^{+}$ and X${}^{-}$. We ascribe this difference to heavy-hole/light-hole (*hh-lh*) mixing [76,77,78,79,80,99]. This feature was first highlighted in DE QDs grown on (001)-oriented substrates and was explained as an effect of the different *hh* and *lh* bands dispersion in the different crystallographic directions [76]. While *e* states can be approximately described as isotropic and parabolic bands, valence *hh* and *lh* bands are affected by strong anisotropy and different curvatures. In spite of the large energy splitting between *hh* and *lh* bands in GaAs (tens of meV), strain and corresponding piezoelectricity [77,79,80,99] (in SK QDs) or shape asymmetries [76,78] (in DE QDs) lead to a substantial mixing of the corresponding *hh* and *lh* states, providing different selection rules (uneven PL intensity) for the two recombination paths. Here, we observe differences in the PL intensity that are of the same order of those observed in the (001) counterpart [76,77,79,80,99].

### 3.4. Line Broadening: Spectral Diffusion

We address the origin of the line broadening by monitoring the line-shape and width of each s-shell excitonic species. By monitoring the line-shape and corresponding broadening of each PL component in the s-shell we observe, respectively, a common Gaussian envelope and a remarkably different full width at half maximum (FWHM; Figure 2b). The common origin of the line broadening in this class of QDs is spectral diffusion: in spite of the expected natural linewidth that should be of the order of a few μeV with a corresponding Lorentian lineshape [82,84,85,96], the measured broadening values (tens to hundreds of μeV) and Gaussian envelopes are the fingerprints of a fluctuating charged environment. Charging and un-charging of electron and holes traps nearby the QDs (within a few tens of nm), produces a variable quantum confined Stark-shift of the excitonic energy levels [24,30,82,83,84,85,86]. A PL measurement lasting for a few seconds acquires many photons emitted at slightly different energy, thus providing a Gaussian lineshape. However, owing to the different *hh* and *e* confinement (larger penetration in the barrier for *e* with respect to *hh*) and a screening of external electric field for many-particles states, the corresponding Stark-shift can be very different. It has been routinely reported a larger broadening of X with respect to the other s-shell excitons, whereas the relative broadening of XX, X${}^{+}$, and X${}^{-}$ can be very different, depending on the carrier trapped and its position with respect to the QD [24,82,95]. As an example, the case in Figure 2b seems to correspond to electrons confined in the plane of the QD [82]. However, this picture can be quite different from a QD to another, owing to the randomness and local changes of the extrinsic disorder.

State-of-the-art droplet epitaxial QDs have been shown capable of extremely narrow PL bands, in the few μeV range [96] and record fidelity in X–XX photon entanglement [27,29] that requires a high degree of coherence. It is also worth mentioning that alternative methods, such as droplet etching, have been proven capable of high quality PL lines with a line broadening of about 20 μeV [16] and entanglement [9]. In this case of (311)A QDs, in spite of the similar high-temperature process used to grow high quality droplet epitaxial nanostructures on (001) substrates, the line broadening is rather large, comparable to lower quality QDs grown at lower temperature [33,82,83,90]. A precise assessment of this phenomenology would require a more systematic study and a clear-cut attribution of the larger defectivity is not possible. We observe that these QDs have a very peculiar shape, with a gap on one side. This feature may lead to a more pronounced tunneling of the excitonic wavefunction in the barrier, where most of the defects are located. Exploring the nearby environment may lead to a more effective interaction with the charged defects that in turn lead to a larger broadening.

### 3.5. Power Dependence: Level Filling

Let us now address the level filling of the s-shell excitons by power dependent measurements (Figure 3). At low power, only two lines, X and X${}^{+}$, are visible with a dominant intensity of X, while when increasing the power also X${}^{-}$ and, soon after, XX appears. At larger power, X and X${}^{+}$ swap their relative intensity and then quench. XX and X${}^{-}$ reach their maximum and quench at very large power. M0 and M1 follows a similar dynamics of X${}^{-}$ and XX, respectively. At lower energy, other peaks appear together with a broad background that then extends to all the spectrum. These latter features are interpreted as the effect of multi-excitons recombining from the p-shell and the coupling with the continuum of states above the confined energy levels. For the sake of thoroughness, we note a slight red-shift of the PL intensity at large excitation power. This effect is interpreted as a heating of the sample. It is also worth noting that the PL intensity of the s-shell excitons can be followed over more than 3 orders of magnitude of the excitation power, meaning that the capture and recombination processes are very efficient in this sample.

A simple but meaningful description of the level filling of individual QDs can be provided by a model based on the assumption that carrier capture and recombination are random processes [100]. By assuming an infinite set of confined levels in the nanostructure, that obviously constitutes a rough approximation, the model can be further simplified to a Poissonian dependence of the PL intensity [81,100]. In spite of its simplicity, this modeling accounts extremely well for power dependence [81] and recombination dynamics [46,101] of the s-shell excitons. In this model, we assume that the PL intensity $I_{PL}$ of a given energy level is proportional to the occupation probability $N_{n}$ of that level:(1)$$I_{PL}\propto N_{n}$$

The Poissonian model for level filling predicts that the occupation of a certain level with *n* excitons $N_{n}$ is provided by
(2)$$N_{n}=\frac{<n{>}^{n}}{n!}e^{-<n>}$$

Thus, for *n* = 1, $N_{1}$ describes the level filling of X, for *n* = 2, $N_{2}$ describes the level filling of XX, etc. <n> is the average number of exciton created in the QD. This number is obtained averaging over many capture/recombination cycles and under steady state excitation and is defined as $<n>=N_{R}\tau_{r}/\tau_{c}$, where $N_{R}$ is the number of e-h pairs created in the reservoir, $\tau_{r}$ is the exciton recombination rate, and $\tau_{c}$ the exciton capture rate. Provided that $\tau_{r}$ and $\tau_{c}$ lie in the ps to ns range and the integration time for a PL measurement last about one second, this hypothesis is well satisfied.

A final phenomenological assumption for <n> that allows to link the experimental data to this Poissonian model is that
(3)$$<n>=\beta P_{exc}^{\alpha }$$
where α and β are constants characterizing the capture mechanism. Note that in the vast majority of works on QDs, the level filling of the s-shell excitons is simply provided by Equation (Equation 3), with α(X)∼1 for X, and α(XX)∼2 for XX. However, this oversimplified assumption does not account for saturation and quenching of the PL lines under CW excitation [31,33,69,102].

The Poissonian model well reproduces the power dependence of X and XX that are extracted by integrating in energy their PL line up to 1600 nW after background subtraction (Figure 3b). Fitting the data using Equations (2) and (3), we obtain $\alpha_{X}=1.1$ (very close to 1 as expected) and a simultaneous fit of X and XX with the same parameters. The saturation power of X is ~1500 nW and about two times larger for XX.

Although the simple Poissonian model allows only for neutral excitons capture (n=1,2,...) and in principle cannot account for charged complexes, a nice fitting of the data can be recovered by assuming half-integer numbers [81]. In this case, the model is modified by adding a phenomenological coefficient γ to $\alpha_{X}$:(4)$$N_{n}=\frac{{\left(<n{>}^{n}\right)}^{\gamma }}{n!}e^{-<n>}$$

Thus, for instance, γ=1 corresponds to a filling dynamics similar to that of the neutral exciton X state, whereas γ=1.5 is *one and an half exciton* (between X and XX), that is, a charged exciton (with an additional spectator charge induced by optical pumping). By using Equation 4 we can nicely fit the power dependence of X${}^{+}$ and X${}^{-}$ by keeping unchanged the parameters α and β that were used to fit X and XX (Figure 3b). We obtain $\alpha \left(X^{+}\right)=1.3\alpha \left(X\right)$ and $\alpha \left(X^{-}\right)=1.6\alpha \left(X\right)$ that are very close to 1.5.

Note that, in this specific QD, the power dependence of $X^{+}$ and $X^{+}$ is superlinear with respect to X. In addition to this, their corresponding saturation power is larger with respect to X. This is not always the case, and a very similar power dependence of charged and neutral complexes can be observed (γ∼1 ) [81]. These differences can be ascribed to the nature of the spectator charge in the QD at the moment of e−h recombination: the presence of the extra carrier can be ascribed either to residual doping of the heterostructure or to optical injection. In the first case, when the extra carrier comes from a donor/acceptor impurity, the corresponding power dependence of the trions follows a trend similar to that one of the neutral counterpart (γ∼1 and same saturation power of *X*) [81]. If instead the spectator carrier is optically injected in the QD, a super-linear trend and a larger saturation power with respect to X is expected. In this latter case, a quasi-Poissonian law with γ>1 well reproduces the experimental data.

We note that also the dynamics of M0 and M1 with incident power can be nicely accounted for by this pseudo-Poissonian model providing γ(M0)=1.5 and γ(M1)=2.1. These values of γ that are larger than 1 account for the super-linear dependence on the incident power and support the multi-exciton nature of these energy levels. Based on the similarities of these lines with those reported in droplet epitaxial QDs grown on (111)A substrates (without a wetting layer as in this case), and on the superlinear power dependence, we speculate that M0 and M1 could be related to recombination of doubly charged excitons [36] or recombination in presence of a carrier in the p-shell [103]. A more conclusive assessment of these lines would require charge tunable devices and magneto-photoluminescence. However, these characterizations go well beyond the aim of this work.

To conclude this section, we estimate the QD capture volume for excitons [81]. This can be deduced from the saturation power of the X level: at that power (of approximately 1500 nW distributed over a diameter of 1 μm) the number of carriers created in the host matrix around the QD provides a steady occupation of the X level by one exciton. Thus, the inverse of the carrier density injected in the semiconductor at that saturation power represents the capture volume. Assuming a penetration depth of the laser light of ~1 μm, an excitonic recombination lifetime of about 1 ns and considering the QD as a sphere, we can roughly deduce a capture radius of the order of 50 nm. This value is rather large with respect to the physical size of the QD, that is of the order of a few nm. We also observe that this value is perfectly in line with those measured in the (001) DE QDs counterpart [81] as well as in other kind of quantum emitters in III–V [95] and in group IV materials [104].

### 3.6. Temperature Dependence

We now address the PL properties of individual QDs as a function of temperature (Figure 4). Bright PL emission can be observed well above liquid nitrogen temperature (Figure 4). A quadratic red-shift of the exciton emission is observed, as expected for this class of nanostructures that follows the empirical Varshini law (Figure 4a). At the same time, a strong quenching of the PL intensity of about two orders of magnitude is observed when passing from 10 to 90 K.

The low-temperature spectrum (up to about 30 K), as discussed in the previous sections, shows sharp lines attributed to the s-shell excitons and is well described by a Gaussian envelope (Figure 2) [82]. At higher temperature instead, when the phonon population plays an important role, we observe the onset of a broad Lorentzian-shaped pedestal below the original sharp line that can be now well approximated with a Lorentzian envelope (Figure 4b) [89,90,105]. The relative intensity of this broad pedestal increases and dominates over the sharp central line. This phenomenology corresponds well to what has been thoroughly explained by using the Huang–Rhys formalism for (001) DE QDs as well as for SK QDs [89,90,105]: the broad band is interpreted as a superposition of acoustic phonon replicas (polaron) whereas the sharp central line emission is assigned to the zero phonon line (ZPL).

## 4. Conclusions

In conclusion, we showed that droplet epitaxial QDs grown on the highly anisotropic (311)A surface are characterized by bright PL lines emitting at visible frequency. These lines correspond well to the picture of s-shell excitons recombination in III–V nanostructures. We measured a relatively large fine structure splitting (10–100 μeV) of the neutral exciton X, that is justified by the anisotropic shape of these QDs. Inhomogeneous line broadening lies at 100 μeV and is ascribed to spectral diffusion originating from the fluctuating charged environment. This broadening is specific of each excitonic species and reflects the nature and position of the trapped carriers. Level filling of the s-shell excitons is well accounted for by a simple Poissonian model that allows to describe the power dependence of the main PL lines and to estimate an excitonic capture volume much larger than the physical size of the QD. Finally, a bright PL emission is still visible well above liquid nitrogen temperature.

Overall, these results account for the relevance of this class of QDs, where X, XX, and X${}^{+}$ emissions can be followed for approximately 3 orders of magnitude of excitation power below saturation and up to relatively large temperature that are appealing features for applications as single-photon sources. The low-temperature linewidth is rather large with respect to state-of-the-art droplet epitaxial quantum dots, and in principle it could be improved by using alternative fabrication processes at higher temperature. Exploiting resonant [85,106,107] or two-photon [29] laser excitation the issues of broadening and spin fine structures can be overcome as shown in similar systems. Another possible way to overcome the large fine structure splitting observed in these QDs for quantum entanglement with X and XX recombination is exploiting the time reordering scheme [95,108,109]. In this case, a negligibly small XX binding energy is required. Although this condition has not yet been reported in (311)A QDs [73], exploiting the lack of a wetting layer (and thus a large lateral confinement) and growing smaller QDs could lead to this special configuration of the energy levels. Finally, spectral selection of entangled photon can be achieved embedding the emitter in a cavity [110] or applying external strain to compensate for the asymmetric quantum dot shape [111].

## Figures and Tables

**Figure 1 nanomaterials-10-01833-f001:**
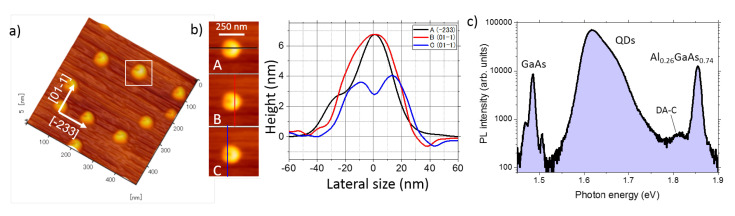
(**a**) Atomic force microscope micrograph representing a 3D view of uncapped GaAs quantam dots (QDs) grown on a (311)A oriented Al${}_{0.26}$Ga${}_{0.74}$As surface. The main crystallographic axes are highlighted. (**b**) Left panels: blow-up of a single QD from panel (**a**). A, B, and C highlight cuts in different crystallographic directions and points of the QD and are reported in the right panel. (**c**) Photoluminescence (PL) spectrum (in semi-log scale) at 5 K showing the emission of (respectively, from the low to the high energy side) the GaAs, the GaAs QDs, the carbon donor–acceptor (DA-C), and the Al${}_{0.26}$Ga${}_{0.74}$As barrier layer.

**Figure 2 nanomaterials-10-01833-f002:**
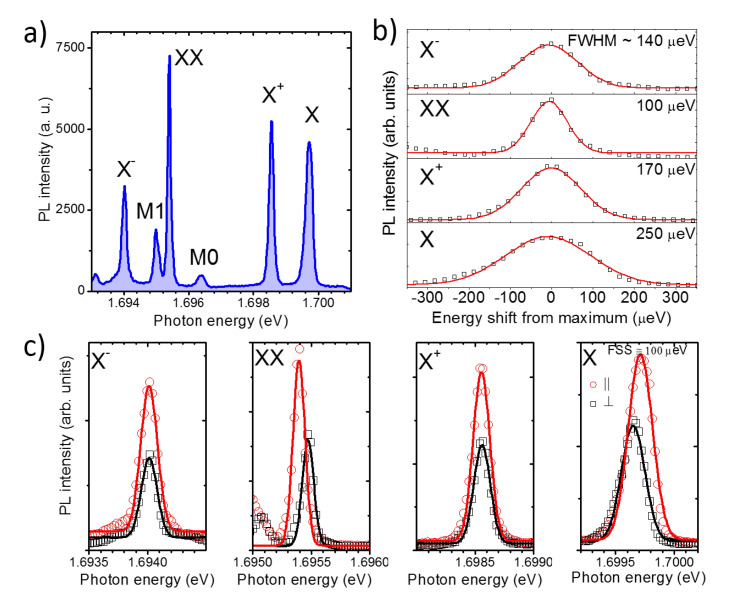
(**a**) PL spectrum of a single GaAs QDs sandwiched between (311)A Al${}_{0.35}$Ga${}_{0.65}$As barrier layers. The labels on the most intense lines X, X${}^{+}$, XX, and X${}^{-}$ highlight, respectively, the emission from the neutral exciton, the positive charged exciton, the neutral biexciton, and the negative charged exciton. M0 and M1 highlight other non-attributed multiexciton complexes. (**b**) Symbols: PL spectra of X, X${}^{+}$, XX, and X${}^{-}$ (respectively, from the bottom to the top panel). Red lines are Gaussian fit to the data. The corresponding full width at half maximum (FWHM) is reported on each panel. (**c**) Linearly polarized components of X, X${}^{+}$, XX, and X${}^{-}$ PL (respectively, from the right to the left panel). Red and black symbols indicate orthogonal polarization. The continuous lines are Gaussian fits. The fine structure splitting (FSS) measured from the X and XX components is ~100 μeV.

**Figure 3 nanomaterials-10-01833-f003:**
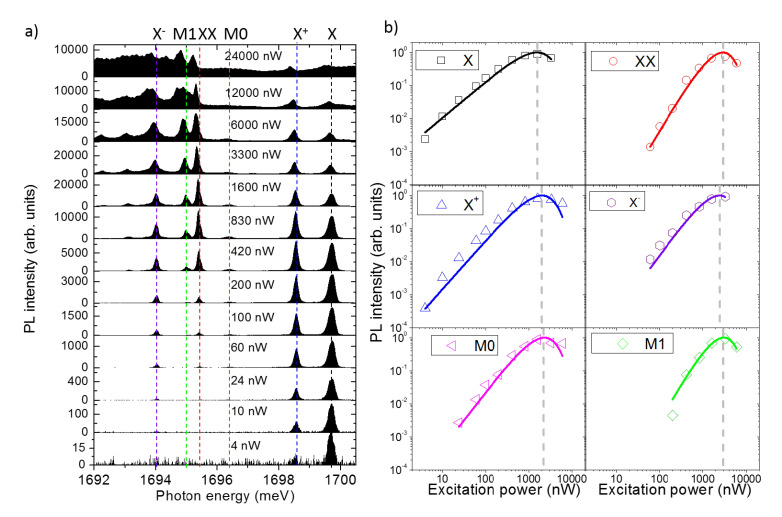
(**a**) PL spectra from low to high excitation power for the same QD shown in Figure 2. The incident power is highlighted on each panel. The vertical dashed lines act as guides for the eyes. (**b**) Evolution of the six main PL lines as a function of the incident excitation power. The PL intensity has been normalized to the maximum for each component. Each panel refers to a specific exciton complex recombination as highlighted by the corresponding labels. Symbols are the experimental data whereas lines are Poissonian fits.

**Figure 4 nanomaterials-10-01833-f004:**
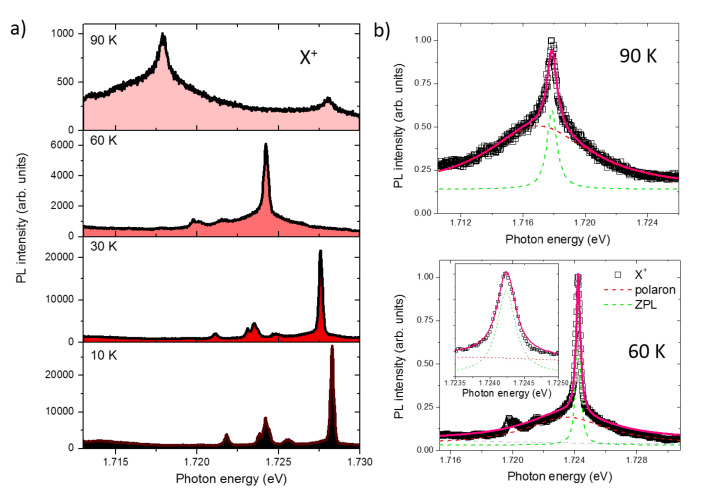
(**a**) Selected PL spectra from 10 to 90 K of a single QD with a bright X${}^{+}$ emission. (**b**) Bottom panel: detail of the spectrum from A at 60 K with Lorentzian fits of the zero phonon line and the polaron band. The inset shows a blow up of the zero phonon line (ZPL). Top panel: detail of the spectrum at 90 K with Lorentzian fits. The measurements have been performed exciting with a laser power of 7000 nW.

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
