# Peer review of "Exciton Dynamics in Droplet Epitaxial Quantum Dots Grown on (311)A-Oriented Substrates"

_nanomaterials, 2020, doi:10.3390/nano10091833_

Round 1

Reviewer 1 Report

Report on

Exciton dynamics in droplet epitaxial quantum dots grown on (311)A oriented substrates

by Marco Abbarchi et al.

The manuscript addresses the optical properties of GaAs QDs formed by droplet epitaxy on (311)A substrates. In my opinion, the paper clearly describes the experiments, the discussion is mostly conclusive, and the topic is of interest. Therefore, I recommend publication after consideration of the points given in the following.

1) From section “2.1. Sample fabrication” and the caption of Figure 1 on might assume that free standing QDs are used for the optical spectroscopy. On the other side, the caption of Figure 2 indicates that the QD are sandwiched between AlGaAs barrier layers. The authors should clarify this.

2) In Figure 1, additional AFM line-scans might be helpful to evaluate the QD shape more quantitatively.

3) For Figure 3a and the corresponding discussion spectra showing also the p-shell emission might be helpful to evaluate the relation between the M0 and M1 peaks and the p-shell occupation.

4) The authors relate the emission linewidth to a crystal trap density. Here a comparison with typical GaAs QDs grown on (001) surfaces with the same MBE system at the same time would be interesting. The broader lines on (311)A could reflect a dot shape more sensitive to charged traps or a stronger incorporation of traps due to the different substrate orientation. 

5) In section “7. Conclusions” the authors state: “Overall, these results account for the high quality …”. This statement is misleading for QDs having rather broad emission lines and a strong fine structure splitting. The authors should clarify, if they claim their QDs for the field of quantum information technology or for applications where a high dot density is advantageous.

6) The references indicates a very high number of self-citations. The authors should check, if really all of their cited papers are directly relevant for the present work.

Reviewer 2 Report

Abbarchi et al. study the optical properties of GaAs/AlGaAs QDs on GaAs(311)A substrates. The groups from Marseille and Tsukuba realized many important works on droplet epitaxy-an epitaxial fabrication method that allows for the realization of strain-free QDs between 600 and 800 nm for single or entangled photon sources. In this manuscript, a complete PL analysis is performed and the effect of the asymmetric QD shape due to anisotropic (311)A surface is investigated. In addition, a Poissonian model well describes the level filling of the excitons. Overall, the results are clearly written and the physics sounds. I recommend the publication in Nanomaterials after the following revision:

- Wang et al., Nanoscale Res. Lett. 1, (2006) 57, Atkinson et al., J. Appl. Phys. 112 (2012) 054303 and  Huo et al., Appl. Phys. Lett. 102 (2013) 152105 are important references to be added at the beginning of the introduction on droplet etching epitaxy (LDE), which is a development of droplet epitaxy. A reference on the strongly entangled QDs obtained by LDE QDs (Huber et al., 8 (2017) 15506) is also missing.

-The authors should comment on the effect of RTA on QD shape. Do they measure any QD high fluctuation? This will also influence the optical properties.

-The worsening of the optical quality could be improved by Al droplet etching of AlGaAs followed by GaAs infilling of the nanoholes (see Huo et al., Appl. Phys. Lett.). In fact, this method allows for defect free QDs and reduction of spectral diffusion. The authors should comment on this in the paper as possible alternative fabrication process.

- The QD spectrum of Fig. 1c shows a peak centered at 1.62 eV. Do the authors measure the same excitonic species in this region as for the spectra measured in Fig. 2a?

- What is the origin of the feature at ~1.8eV in Fig. 1c?      

-In Fig. 3a the quenching of the intensity is reported and it occurs at lower power for X than for X+. The authors should comment on this.

- The physical size of the QD is not clear from the manuscript. The authors should give more details.

 Others:

  • The scan size of Fig. 1b should be added;
  • X- and XX, respectively (157) should be read X+ and X-, respectively.
  • Power value in Fig. 4 is missing.

Reviewer 3 Report

Dear Editor,

I accurately reviewed the article

Manuscript Number: nanomaterials-901954-v1

Title: Exciton dynamics in droplet epitaxial quantum dots grown on (311)A oriented substrates

submitted to Nanomaterials

In this article quantum dots grown on (311)A-oriented surface are obtained with record surface density, with or without a wetting layer. Authors address the main optical and electronic properties of s-shell excitons in individual quantum dots grown on (311)A substrates with photoluminescence spectroscopy experiments. They show the presence of neutral exciton and biexciton as well as positive and negative charged excitons. They investigate the origins of spectral broadening, identifying them in spectral diffusion at low temperature and phonon-interaction at higher temperature, the presence of fine interactions between electron and hole spin, and a relevant heavy-hole/light-hole mixing. They interpret the level filling with a simple Poissonian model reproducing the power excitation dependence of the s-shell excitons. These results are relevant for the further improvement of this class of quantum emitters and their exploitation as single photon sources for low density samples as well as for efficient lasers for high density samples.

The topic is interesting, but the authors have to solve some issues.

Introduction

In the introduction would be useful for readers an overview to quantum dots, their several preparation and applications, such as for example:

  1. State-of-the-Art and Trends in Synthesis, Properties, and Application of Quantum Dots-Based Nanomaterials; Part. Part. Syst. Charact. 2019, 36, 1800302
  2. Sensitivity to heavy-metal ions of open-cage fullerene quantum dots; Sensors 17 (2017) 2614 doi:10.3390/s17112614
  3. Fabrication and full characterization of state-of-the-art quantum dot luminescent solar concentrators; Solar Energy Materials and Solar Cells Volume 95, Issue 8, August 2011, Pages 2087-2094
  4. Nucleobases functionalized quantum dots and gold nanoparticles bioconjugates as a FRET system: synthesis, characterization and potential applications; J. Colloid Interf. Sci. 514 (2018) 479-490
  5. Advances in carbon dots: from the perspective of traditional quantum dots; Mater. Chem. Front., 2020,4, 1586-1613

Experimental part

The experimental part lacks information on AFM measurements, the instrument used and statistics.

Results and discussion

In results and discussion it would be good to start with an introductory sentence for figure 1.

The paragraphs indicated with 4 and following should be inserted as a sub-paragraph in results and discussion, introducing the first figure with introductory sentences

References should be more recent, 19 out of 64 are prior to 2010

Furthermore, the completeness and editing of various references must be checked

English requires some corrections, there are some typos and many sentences are too long.

In conclusion, the article is suitable for publication in Nanomaterials after minor revisions.

best regards
